# The Influence of Race/Ethnicity on the Transcriptomic Landscape of Uterine Fibroids

**DOI:** 10.3390/ijms241713441

**Published:** 2023-08-30

**Authors:** Tsai-Der Chuang, Nhu Ton, Shawn Rysling, Derek Quintanilla, Drake Boos, Jianjun Gao, Hayden McSwiggin, Wei Yan, Omid Khorram

**Affiliations:** 1Department of Obstetrics and Gynecology, Harbor-UCLA Medical Center, Torrance, CA 90502, USA; tchuang@lundquist.org; 2The Lundquist Institute for Biomedical Innovation, Torrance, CA 90502, USA; nhu.ton@lundquist.org (N.T.); shawn.rysling@lundquist.org (S.R.); derek.quintanilla@lundquist.org (D.Q.); drake.boos@lundquist.org (D.B.); gaojianjunxf@gmail.com (J.G.); hayden.mcswiggin@lundquist.org (H.M.); wei.yan@lundquist.org (W.Y.); 3Department of Medicine, David Geffen School of Medicine at University of California, Los Angeles, CA 90095, USA; 4Department of Obstetrics and Gynecology, David Geffen School of Medicine at the University of California, Los Angeles, CA 90095, USA

**Keywords:** fibroid, race, ethnicity, MED12 mutation, next generation RNA sequencing

## Abstract

The objective of this study was to determine if the aberrant expression of select genes could form the basis for the racial disparity in fibroid characteristics. The next-generation RNA sequencing results were analyzed as fold change [leiomyomas/paired myometrium, also known as differential expression (DF)], comparing specimens from White (n = 7) and Black (n = 12) patients. The analysis indicated that 95 genes were minimally changed in tumors from White (DF ≈ 1) but were significantly altered by more than 1.5-fold (up or down) in Black patients. Twenty-one novel genes were selected for confirmation in 69 paired fibroids by qRT-PCR. Among these 21, coding of transcripts for the differential expression of *FRAT2*, *SOX4*, *TNFRSF19*, *ACP7*, *GRIP1*, *IRS4*, *PLEKHG4B*, *PGR*, *COL24A1*, *KRT17*, *MMP17*, *SLN*, *CCDC177*, *FUT2*, *MYO5B*, *MYOG*, *ZNF703*, *CDC25A*, and *CDCA7* was significantly higher, while the expression of *DAB2* and *CAV2* was significantly lower in tumors from Black or Hispanic patients compared with tumors from White patients. Western blot analysis revealed a greater differential expression of PGR-A and total progesterone (PGR-A and PGR-B) in tumors from Black compared with tumors from White patients. Collectively, we identified a set of genes uniquely expressed in a race/ethnicity-dependent manner, which could form the underlying mechanisms for the racial disparity in fibroids and their associated symptoms.

## 1. Introduction

Uterine fibroids are benign tumors that disproportionately affect Black over White and Hispanic women in terms of prevalence, symptom severity, size and number of tumors, and age of onset [1,2,3,4]. This racial disparity exerts a significant impact on the quality of life for patients with either benign [5] or malignant tumors [6]. Because of the disease burden in Black women, they have less access to minimally invasive surgery and experience higher rates of complications with hysterectomies [7]. Several mechanisms have been proposed for this racial disparity in fibroid characteristics and associated symptomology [8,9], including a deficiency in vitamin D levels [10], which is known to play a role in fibroid pathogenesis [11], and the presence of expanded myometrial stem cells in Black compared with those of White women [12]. Several race-dependent gene polymorphisms have also been associated with fibroids, including genes involved in estrogen synthesis, e.g., Cyp17 [13], estrogen metabolism, e.g., Catechol-O-methyltransferase (COMT) gene, which is involved in catechol estrogen synthesis [14,15,16], and in the genes regulating the retinoic acid pathway [1]. Earlier studies also showed greater expression of aromatase genes in tumors from Black women, compared with genes of White women, which provide a local source of estrogen known to stimulate fibroid growth [17]. We recently reported on a marked dysregulation of tryptophan catabolism in fibroids in Black, White, and Hispanic women but with greater dysregulation of TDO2 and other enzymes in the tryptophan metabolism in Black women [18,19]. In another publication, we reported on the differential expression of super-enhancer lncRNAs (SE-lncRNAs) and their associated mRNAs in leiomyoma and the matched myometrium. We found that a number of the SE-lncRNAs/mRNA pairs (SOCS2-AS1/SOCS2, RP11-353N14.2/CBX4, RP1-170O19.14/HOXA11, and RP11-225H22/NEURL1) were significantly higher in tumors from Black patients compared with those of White patients [20]. Genome-wide association studies have revealed an association of a number of genes with fibroid size and volume in a race-dependent manner [21,22].

Several gene profiling studies have provided evidence for aberrant expression of a number of genes in fibroid tumors of Black patients by microarray [23] and next-generation sequencing [24,25]. Studies have also identified a number of pathways that are differentially affected, including the TGF-β pathway [23], reactive oxygen species, hypoxia, and oxidative phosphorylation pathways [25]. In the most recent profiling studies, the myometrial race-dependent expression of genes was also evaluated [24,25]. The authors in this study [24] suggested that the race-dependent aberrant expression of genes could form the basis for increased susceptibility to develop fibroids. There was no concordance among these three gene profiling studies [21,22,23] regarding genes selectively altered in expression in a particular race. One of these earlier studies [23] concluded that no race-specific genes are misexpressed in fibroids, and only the relative degree of change in expression is race-dependent. Our objective in this gene expression profiling study was, first, to determine if there are genes that are uniquely expressed in a race-dependent manner and, second, to confirm these findings from NGS in a large sample set by qRT-PCR. We hypothesized that a unique set of genes are aberrantly and differentially expressed in fibroids from Black women than in White women, which could provide a mechanism for differences in race-associated tumor characteristics. 

## 2. Results

### 2.1. Differential Expression of Race/Ethnicity-Associated Coding RNA Transcripts in Leiomyoma and Matched Myometrium

We previously reported on the effect of MED12 mutation on the expression of genes in fibroids [26] and identified 394 genes that were aberrantly and differentially expressed in mutated tumors. Of 31 novel coding transcripts confirmed by qRT-PCR [26], the expression of several, including *EZH2*, *EGFL6*, *ITGA9*, *WNT4*, *WNT2*, *WNT16*, *PPP1R14C*, *ATP5MC1P1*, *CACNA1D*, and *BMP7*, also correlated with race/ethnicity (Appendix A). Among these genes, *ATP5MC1P1* was minimally altered in the White group (Leiomyoma/paired Myometrium) while significantly increased in the Black group (Appendix A). Further race/ethnicity-dependent comparisons for these genes in leiomyoma and myometrium are shown in Appendix A.

In the present profiling study, we found a significant overlap of differentially expressed genes between the Black group and White group. However, in the Black group, the degree of change was greater for most genes. Following the normalization of 29354 RNA transcripts, the analysis was performed as fold change (Leiomyoma/paired Myometrium) comparing the Black with White group. This analysis based on differential expression resulted in the identification of 3819 RNA transcripts with altered expression, of which the expression of 1510 RNA transcripts was increased, while the expression of 2309 RNA transcripts was decreased by 1.5-fold or greater in the Black group compared with the White group. 

Hierarchical clustering and TreeView analysis separated these transcripts into their respective groups (Figure 1A). We identified 95 transcripts that showed more than 1.5-fold change (up or down) in the Black group but not in the White group (Table 1). The heat map (Figure 1B) shows the 95 transcripts that were uniquely altered in expression in the Black group. The Gene Ontology (GO) and KEGG (Kyoto Encyclopedia of Genes and Genomes) pathway enrichment analysis for these 95 transcripts revealed that the genes that were uniquely expressed in the Black group were predominantly involved in the regulation of cell signaling and extracellular matrix (ECM) constituents (Figure 2A). Using the STRING database and Cytoscape software version 3.9.1 the functional analysis of the protein-protein interactions (PPI) network was constructed, demonstrating the highlighted association between target genes that are involved in fibroid pathogenesis (Figure 2B). The involvement of many proteins, such as LHFPL3, SGK1, and RhoA, shown in the protein-protein interaction network (Figure 2B), is well established in fibroid pathogenesis [27,28,29,30]. Other proteins, such as CDCA7, CAV2, MPST, CCDC6, GBP3, and DDR2, are novel and require further investigation.

### 2.2. Validation of Race/Ethnicity-Associated Coding RNA Transcripts in Leiomyoma and Matched Myometrium

To provide support for and validate the NGS data, we selected differentially-expressed coding transcripts for confirmation studies by qRT-PCR in a large sample set (n = 69), including the same tissues that were used for RNAseq. Among the 21 genes for validation, seven (CAV2, CDCA7, DAB2, GRIP1, MMP17, MYO5B, and ZNF703) were selected from the 95 genes that were altered significantly in tumors from the Black group but minimally changed in White patients. The remaining 14 genes were selected from our RNAseq analysis because of their significant race dependence and their functional involvement in reproduction. The Gene Ontology (GO) and KEGG (Kyoto Encyclopedia of Genes and Genomes) pathway enrichment analysis for this group of differentially-expressed RNA transcripts revealed that these genes were involved in multiple signaling pathways, such as signal transduction (*FRAT2*, *SOX4*, *TNFRSF19*, *ACP7*, *GRIP1*, *IRS4*, *PLEKHG4B*, and *PGR*), ECM organization (*COL24A1*, *KRT17*, and *MMP17*), metabolism of proteins (*CCDC177*, *FUT2* and *MYO5B*), ion homeostasis (*SLN*), transcription regulation (*MYOG* and *ZNF703*), and regulation of cell proliferation (*CDC25A*, *DAB2*, *CAV2* and *CDCA7*), which are all relevant to leiomyoma pathogenesis. Among the 21 coding transcripts validated the expression of *FRAT2*, *SOX4*, *TNFRSF19*, *ACP7*, *GRIP1*, *IRS4*, *PLEKHG4B*, *PGR*, *COL24A1*, *KRT17*, *MMP17*, *SLN*, *CCDC177*, *FUT2*, *MYO5B*, *MYOG*, *ZNF703*, *CDC25A* and *CDCA7* was significantly higher, while the expression of *DAB2* and *CAV2* was significantly lower in leiomyomas as compared to matched myometrium in the combine race/ethnicity groups (Appendix A). The 21 differentially expressed transcripts (Leiomyoma/paired Myometrium) were further broken down by race/ethnicity (Figure 3). In this analysis the expression of *FRAT2*, *SOX4*, *TNFRSF19*, *ACP7*, *GRIP1*, *IRS4*, *PLEKHG4B*, *PGR*, *COL24A1*, *KRT17*, *MMP17*, *SLN*, *CCDC177*, *FUT2*, *MYO5B*, *MYOG*, *ZNF703*, *CDC25A* and *CDCA7* was significantly higher, while the expression of *DAB2* was significantly lower in Black group as compared to White group. In addition, the expression of *CAV2* mRNA was significantly lower in tumors from Hispanic patients than in tumors from White patients (Figure 3). Our analysis showed nine transcripts (*FRAT2*, *TNFRSF19*, *GRIP1*, *PGR*, *KRT17*, *SLN*, *CDC25A*, *FUT2*, and *ZNF703*) were minimally or not altered in expression in the White group (Leiomyoma/paired Myometrium ≈ 1) while significantly higher in tumors from the Black group (Figure 3). Further comparisons among the three race/ethnicity groups were made based only on expression levels in the fibroid and in the myometrium (Figure 4). A comparison of myometrial expression of genes revealed significant race-related differences in expression for *FRAT2*, *ACP7*, *GRIP1*, *KRT17*, *SLN*, *MYO5B*, *MYOG*, and *CDCA7* (Figure 4). For comparison of differential expression of genes in leiomyomas, there was a significant race/ethnicity correlation in expression for *TNFRSF19*, *IRS4*, *PLEKHG4B*, *PGR*, *KRT17*, *CCDC177*, *MYO5B*, and *ZNF703* (Figure 4). Among the 21 transcripts with PCR confirmation, the expression of *FRAT2*, *TNFRSF19*, *ACP7*, *IRS4*, *PLEKHG4B*, *KRT17*, *ZNF703*, and *CAV2* was significantly higher in MED12-mutation-positive specimens as compared to MED12-mutation-negative ones (Leiomyoma/paired Myometrium; Appendix A). A summary of the analysis from Figure 3 and Figure 4 and Appendix A is shown in Table 2.

### 2.3. Validation of PGR Protein Expression in Leiomyoma and Matched Myometrium

Because of the pivotal role of progesterone in fibroid pathogenesis [54] and our results indicating a race/ethnicity effect on PGR mRNA expression, we carried out additional analysis for PGR protein expression by immunoblotting with the same paired specimens used for qRT-PCR analysis, including nine paired specimens from White patients, 23 from Black, and 24 from Hispanic. As shown in Figure 5, total PGR (A&B) and PGR-A expression were significantly higher in fibroids than in the matched myometrium (Figure 5B), and this effect was race-dependent, with higher protein levels in Black than in White patients (Figure 5C).

## 3. Discussion

The results of this study demonstrate race-dependent differences in the transcriptome of uterine fibroids. Using next-generation sequencing, we identified 95 genes that were differentially expressed (L/M) in specimens from Black but not White patients. We confirmed the expression of 21 genes by qRT-PCR in a large sample set (n = 69), and this analysis also included specimens from Hispanic patients. Of these qPCR-confirmed genes, the differential expression of nine genes (*FRAT2*, *TNFRSF19*, *GRIP1*, *PGR*, *KRT17*, *SLN*, *CDC25A*, *FUT2*, and *ZNF703*) was minimally altered in fibroids matched to myometrium from White patients while significantly higher in expression in tumors from Black patients and less so in Hispanic patients. Two genes (*CAV2* and *DAB2*) confirmed by qPCR showed a decreased differential expression in tumors from Black or Hispanic compared with tumors from White patients. A comparison of myometrial expression of genes revealed significant race-related differences in expression for *FRAT2*, *ACP7*, *GRIP1*, *KRT17*, *SLN*, *MYO5B*, *MYOG*, and *CDCA7*. The expression of several race-associated genes was also dependent on the MED12 mutation status of the tumor, being higher (*FRAT2*, *TNFRSF19*, *ACP7*, *IRS4*, *PLEKHG4B*, *KRT17*, *SLN*, and *ZNF703*) or lower (*CAV2*) in the mutated specimens compared with wild-type tumors. Broadly, the race-associated fibroid genes identified fell mainly in the category of cell proliferation/cell cycle, cytoskeleton regulation, and the important pathways known to be pivotal in fibroid pathogenesis, namely the WNT/β-catenin pathway, the Rho-ROCK pathway, and inflammation. In general, the magnitude of fold change (L/M) in gene expression was most profound in tumors from Black patients, less profound in Hispanic patients, and least profound in White patients. The presence of MED12 mutation additionally added to the effect of race on gene expression. These data provide a molecular basis for the excessive growth and rapid progression of fibroids and consequent associated symptoms in Black women.

One of the hallmarks of the racial disparity in fibroid symptomology is the excessive number, size, and growth progression of tumors in Black patients [1,2], suggesting that race influences the expression of genes that regulate the cell cycle. Our group [55] and others [11] previously reported on aberrant expression of cell cycle regulatory proteins in fibroids, although race as a variable was not taken into consideration in these prior studies. In the current study, the differential mRNA expression of a number of genes (*CDC25A*, *FRAT2*, *FUT2*, *MYOG*, *ZNF703*, *TNFRSF19*) regulating the cell cycle and apoptosis in fibroids from White patients was altered minimally. Still, it was significantly higher in tumors from Black patients and somewhat higher in Hispanics. The aberrant expression of these genes could account for the race-dependent differences in fibroid cell proliferation and growth. We identified other genes with a role in cell cycle control, which were overexpressed in all races, but the magnitude of overexpression was higher in tumors from Black patients. These include *SOX4*, *IRS4*, and *CDCA7*. One of the race-associated, uniquely expressed cell cycle regulatory proteins was CDC25A. This phosphatase removes the inhibitory phosphorylation in several cyclin-dependent kinases (CDKs), including CDK2/CDK1 [56], CDK4 [57], and CDK6 [58], and positively regulates their activity, leading to G1/S and G2/M cell cycle progression. There are three E2F binding regions in CDC25A gene which activate its transcription [59], and its overexpression in different cell types induces tumorigenesis [60]. CDC25A is also a regulator of apoptosis, functioning as a suppressor when it is in the cytoplasm, where it is mostly localized, and as a stimulator of apoptosis when it is in the nucleus [60]. Transgenic overexpression of CDC25A has also been shown to induce tumorigenesis by inducing DNA damage [61]. Another uniquely expressed cell cycle regulatory protein is ZNF703, which is overexpressed in ER/PR-positive breast cancer [62,63] and promotes tumor cell proliferation and invasion in colorectal cancer [64]. Knockdown of ZNF703 in triple-negative breast cancer cells was shown to inhibit the expression of cyclin D1, CDK4, CDK6, and E2F1 and upregulation of Rb1 [63]. FUT2 also plays a role in cell proliferation, and its knockdown inhibited cell proliferation in bovine venular endothelial cells, malignant human A431 cells [65], and breast cancer cells [66]. The tumorigenic effect of FUT2 in colorectal cells was shown to be via the WNT/β-catenin pathway [46]. TNFRSF19, a member of the TNF receptor superfamily, was another gene minimally altered in fibroids from matched myometrium in tumors from White patients but significantly overexpressed in tumors from Black and Hispanic patients. TNFRSF19 is overexpressed in glioblastoma correlating with tumor grade [67], in myeloproliferative neoplasms [68], melanoma [69], lung cancer [70], and nasopharyngeal carcinoma where it is required for cell proliferation [34]. Of the cell cycle genes that were overexpressed in all races but more so in the Blacks was IRS4, which has a stimulatory effect on cell proliferation [71,72,73]. There was elevated nuclear expression of IRS4 in liver cancer and treatment of HEPG2 cell, a liver cancer cell line with IGF-1 and EGF-induced nuclear translocation of IRS4, stimulating cell proliferation by a β-catenin/Rb/cyclin D mechanism [74]. IRS4 is overexpressed in colorectal cancer, and when overexpressed in RKO cells, it induces the expression of cyclin D1, Cyclin E, E2F1, and pRB Ser 705 and pRB Ser809/811 genes [75]. SOX4 is a transcription factor with increased expression in a number of malignancies where its expression is positively correlated with increased cell proliferation, inhibition of apoptosis, and tumor progression [76]. Two genes upregulated in tumors from all races and with a role in tumorigenesis were PPP1R14C, which was shown to promote breast cancer cell proliferation, maintaining GSK3 in an inactive state in triple-negative breast cancer [77], and EGFL6, which regulates angiogenesis and tumor growth [78,79]. Two of the genes we confirmed by qPCR showed a race-dependent decrease in expression, including *DAB2* and *CAV2*. CAV2 is a major component of caveolae and is involved in signal transduction, cell growth, and apoptosis [80]. Additionally, CAV2 has been mapped to a known tumor suppressor locus [81] within a fragile site that is frequently deleted in human cancers [82] and thus may have a tumor suppressor role. DAB2 may also have a tumor suppressor role as it is normally expressed in ovarian epithelial cells but downregulated or absent in ovarian cancer cell lines [83]. This gene shows growth-inhibitory activity in prostate cancer [84]. Overall, our data indicate an overexpression of genes that support cell proliferation and a decrease in the expression of tumor suppressor genes in fibroids from Black compared to those of White patients, with the expression levels in Hispanics being in between those of the Black and White groups.

The next group of genes (*KRT17*, *COL24A1*, *MMP17*, *MYOG*, *MYO5B*, and *GRIP1*) with minimal to no alteration in matched myometrium in the White patients but with significantly higher expression in fibroids from Black and Hispanic patients were related to the cell cytoskeleton and ECM. One of the hallmark characteristics of fibroid tumors is the excess ECM production, which contributes to its stiffness [85,86,87]. The ECM is a reservoir of growth factors and cytokines, and changes in ECM stiffness result in signals that are converted into chemical changes [85]. KRT17 is a type of intermediate filament and part of the cell cytoskeleton. It regulates cell proliferation and skin inflammation, and it is aberrantly overexpressed in a number of malignancies, including breast, cervical, gastric, and oral squamous carcinoma [88]. DNA damage, which is commonly found in fibroids [89], is known to induce the expression of KRT17 [90]. This protein also binds to NF-kB to induce the expression of inflammatory genes [91] and induces angiogenesis [92]. Our previous study [93,94] and others [95] have reported on the significance of NF-kB activation and inflammatory pathways in fibroid pathogenesis. The race-associated aberrant expression of KRT17 could thus be of great relevance to fibroid pathogenesis and merits further investigation. MMP17 is a protease anchored to the membrane and involved in ECM remodeling in physiologic processes such as tissue regeneration/repair and inflammation [43], but also involved in pathologic processes of cancer where it may play a role in cancer progression [96,97,98]. COL24A1 is an ECM component shown previously to be overexpressed in hepatocellular carcinoma [99] and head and neck cancer [100]; in both malignancies, its expression correlated with poor prognosis. Proteomic analysis showed COL24A1 was one of the most abundantly expressed proteins in small-sized fibroids [101]. GRIP1 is a scaffold protein that mediates the trafficking and membrane organization of several transmembrane proteins [37] and is required for normal cell-matrix interaction [102]. Two other structural proteins showing increased differential expression in fibroids from Black and Hispanic patients were MYOG and MYO5B. MYOG, or myogenin, is a basic helix-loo-helix transcription activator that induces myogenesis [103] and is expressed in myofibroblasts of the liver and kidney [104], raising the intriguing potential for a role of this gene in inducing the differentiation of fibroblast into myofibroblast and giving them contractile property in fibroids from Black patients. MYO5B has a role in mitosis, cell polarity [105,106], and tumorigenesis [107]. Of note is that this protein has been shown to facilitate FN1 secretion from human pleural mesothelial cells [108], and excess FN1 expression is a hallmark of fibroids [26,109,110]. Other genes related to cell cytoskeleton and ECM that were overexpressed in tumors from all races, but more so in Black patients, were ITGA9 and BMP7. ITGA9 which is one of the integrin subunits and is aberrantly expressed in many tumors [111]. ITGA9 functions as a receptor of adhesion molecules and ECM and is involved in cell proliferation, adhesion, and angiogenesis. BMP7 is a member of the TGF superfamily. In kidneys, BMP7 has an anti-fibrotic effect inducing ECM degradation [112] and is similarly anti-fibrotic in the liver [113]. BMP7 synergizes with TGF-β1 in the bone to promote ECM synthesis [114] and is anti-fibrotic in the asthmatic lung through its antagonist effect on TGF-β1 [115]. BMP7 has anti-inflammatory effects on the heart [116].

A significant amount of evidence has established the role of WNT/β-catenin in fibroid pathogenesis [20,26,117]. Our profiling indicated a number of proteins in this pathway and showed greater differential expression in tumors from Black and Hispanic patients than in those from White patients. Among these genes, FRAT2, which belongs to the GSK3 binding protein family, is a positive stimulator of the WNT/β-catenin pathway [118,119]. Other genes, including WNT2, WNT4, and WNT16, although elevated in all three groups, were significantly higher in tumors from Black patients. These results indicate potentially greater activation of the WNT/β-catenin pathway in fibroids from Black patients, which could promote more rapid growth of tumors.

Another pathway activated in fibroids is the Rho-ROCK pathway [85,120,121]. One of the fibroid genes we identified, which was markedly overexpressed in all races but more so in Blacks, was PLEKHG4B, which is a guanyl nucleotide exchange factor. This protein catalyzes the conversion of Rho protein from its inactive state to its active state [122] and plays a role in actin cytoskeleton remodeling [39].

Two other genes found to be overexpressed in tumors from all races but more so in Blacks were EZH2 and ACP7. Fibroids are characterized by epigenetic gene modifications [89], and several previous studies have shown overexpression of EZH2, an epigenetic enzyme in fibroids [26,123,124]. EZH2 is a histone-lysine N-methyltransferase enzyme that participates in histone methylation and induces transcriptional repression. Here, we report greater differential expression of EZH2 in tumors from Black patients than in those from White patients, indicating greater epigenetic gene modification in this group. ACP7 is an acid phosphatase with marked relative expression in fibroids in all groups but more so in Blacks; the physiologic function of this gene is unexplored, except in one study showing an association with glioblastoma survival [36].

Of great significance to fibroid biology was the novel finding of a race-associated increase in the differential (L/M) expression of PGR in Black patients but not White patients, with no race-associated differences in myometrial expression of this gene. The mitogenic role of progesterone in fibroids and its stimulatory effect on ECM production is well-established [125]. Furthermore, increased expression of PGR in fibroids as compared with matched myometrium has been previously reported [126,127]. A previous study by immunohistochemistry also indicated a significant increase of PGR-A in Black compared to White patients [128]. Our results indicating race-dependent differences in PGR expression suggest greater responsiveness of tumors from Black patients to progesterone, which could promote greater growth and progression of these tumors in this population. This finding raises the possibility that the common use of progesterone in clinical practice to mitigate symptoms associated with fibroids could potentially be more detrimental in Black patients.

The MED12 mutation rate of specimens used in this study was 55.6% in White, 73.9% in Black, and 62.2% in the Hispanic groups. Although there was a slightly higher MED12 mutation rate in the Black group, among the nine genes (*FRAT2*, *TNFRSF19*, *GRIP1*, *PGR*, *KRT17*, *SLN*, *CDC25A*, *FUT2*, and *ZNF703*) that were minimally or not altered in the White group but significantly more expressed in the Black group, the expression of only four genes (FRAT2, TNFRSF19, KRT17, and ZNF703) correlated with MED12 mutation. In addition, among the 21 genes that were confirmed to exhibit racial differences in their expression, only eight genes (*FRAT2*, *ACP7*, *TNFRSF19*, *IRS4*, *PLEKHG4B*, *KRT17*, *ZNF703*, and *CAV2*) correlated with MED12 mutation status. Furthermore, several reports [129,130,131,132] have indicated that the MED12 mutation rate in leiomyomas is about the same regardless of race/ethnicity. Therefore, we consider race as an important factor leading to differential gene expression, and the combination of positive MED12 mutation status and Black race could magnify the degree of aberrant expression of genes regulating the growth of tumors. However, we could not rule out the impact of MED12 mutation status in our racial analysis because of our limited number of specimens in each race/ethnicity group.

Recent studies aimed at deciphering the pathogenesis of fibroid and the underlying mechanism for aberrant gene expression have stressed the analysis of the myometrial compartment as potentially the source of the tumor onset [24]. In our profiling, we found that the comparison of gene expression in the myometrium showed an opposite expression pattern to results compared as differential expression (Leiomyoma/paired Myometrium). This finding applied to *ACP7*, *GRIP1*, *KRT17*, *SLN*, *MYO5B*, *MYOG*, and *CDCA7*, where mRNA levels were lower in the myometrium in specimens from Black patients compared to myometrium in specimens from White patients, with no differences in expression of these genes comparing specimens from White and Hispanic patients. This finding suggests a potential communication between fibroids and adjacent myometrium, which could influence gene expression.

Three other studies have addressed the role of race on fibroid gene expression using microarray [23] or next-generation sequencing [24,25]. There is no consensus among these studies or with the current study on genes that are uniquely dysregulated in tumors from Black patients compared to those of White patients. This lack of consensus may result in part from a lack of critical validation of RNAseq data by a quantitative technique such as qRT-PCR or immunoblotting in two of the studies [24,25]. Our profiling identified VWF and BDNF as dysregulated genes in fibroids. We validated our study by qRT-PCR for VWF; in contrast to the RNAseq data of Paul et al. [24], we found decreased expression of VWF in fibroids compared to its matched myometrium and did not find any race-related differences in the myometrial expression of this gene. In contrast to a recent report by Li et al. [25], our RNAseq data did not identify any significant race-dependent differences in the expression of reported genes associated with ROS generation, hypoxia, and oxidative phosphorylation. The differences among the profiling studies could be partly due to differences in the technique used for evaluating gene expression and differences in validation studies or lack thereof of such analysis. A strength of our study is that we matched myometrium for comparison for all specimens used in contrast to two other studies [24,25], and we validated the RNAseq by PCR in a large sample set (n = 69). A limitation of our study was a relatively small number of specimens from White patients, which reflects the study population at our institution. Therefore, it is critical that our findings for genes with relative expression of one in the White population be confirmed in a large sample set.

In summary, we have identified a unique set of genes that are aberrantly expressed in fibroids from Black patients compared to tumors from White patients. Dysregulation of these genes could provide a mechanism for race-dependent differences in fibroid characteristics and symptomology. These signature genes predominantly were related to the regulation of cell proliferation and apoptosis, cell cytoskeleton, ECM composition, and inflammation and were involved in the WNT/β-catenin Rho-ROCK pathway, all of which are relevant and critical in fibroid pathogenesis. The presence of MED12 mutation is an additive factor in considering Black race as a factor associated with aberrant gene expression in fibroids. Additional functional studies will be needed to establish the functional significance of these race-dependent, aberrantly expressed genes and how they contribute to the racial disparity in fibroids.

## 4. Materials and Methods

### 4.1. Myometrium and Leiomyoma Tissues Collection

To reduce the variance among fibroids, tumors 3 to 5 cm in diameter and with intramural location (n = 69) were obtained from patients at Harbor-UCLA Medical Center undergoing hysterectomy. Prior approval from the Institutional Review Board (18CR-31752-01R) at the Lundquist Institute was obtained. Informed consent was obtained from all the patients participating in the study who were not taking any hormonal medications for at least three months before surgery. The paired tissues were from White (Caucasians; n = 9), Black (African Americans; n = 23), and Hispanic (n = 37) women aged 30–54 years (mean 45 ± 5.7 years). As previously described, the tissues were snap-frozen and stored in liquid nitrogen for further analysis [133,134].

### 4.2. MED12 Mutation Analysis

Genomic DNA from leiomyomas and paired myometrial specimens was extracted from 100 mg of freshly frozen tissue using MagaZorb DNA Mini-Prep Kit (Promega, Madison, WI, USA) according to the manufacturer’s protocol. PCR amplification and Sanger sequencing (Laragen Inc., Culver City, CA, USA) were performed to investigate the MED12 exon two mutations using the primer sequences in the 5′–3′ direction: sense, GCCCTTTCACCTTGTTCCTT and antisense, TGTCCCTATAAGTCTTCCCAACC. PCR products were sequenced using Big Dye Terminator v.3.1 sequencing chemistry. The sequences were analyzed with the Software ChromasPro 2.1.8 and compared with the MED12 reference sequence (NG_012808 and NM_005120). The 19 pairs of tissues used for next-generation RNA sequencing were from 11 MED12 mutation-positive and eight MED12 mutation-negative leiomyomas. The mutation analysis of the specimens (n = 69) indicated that 46 leiomyomas had the MED12 mutations (46/69 pairs; 66.7%) with no mutations in the myometrium. Missense mutations in MED12 exon 2 were the most frequent alteration (37/46 pairs), followed by in-frame insertion-deletion type mutations (9/46 pairs). The missense mutations in exon 2 included c.130G>C (p.Gly44Arg) (5/37 pairs); c.130G>A (p.Gly44Ser) (7/37 pairs); c.130G>T (p.Gly44Cys) (2/37 pairs); c.131G>C (p.Gly44Ala) (2/37 pairs); c.131G>A (p.Gly44Asp) (15/37 pairs); c.131G>T (p.Gly44Val) (5/37 pairs); and c.128A>C (p.Gln43Pro) (1/37 pairs). The MED12 mutation rate in White patients was 55.6% (5/9 pairs), 73.9% in Black (17/23 pairs), and 62.2% in the Hispanic group (23/37 pairs).

### 4.3. RNA Sequencing and Bioinformatic Analysis

Total RNA was extracted from leiomyoma and matched myometrium using TRIzol (Thermo Fisher Scientific Inc., Waltham, MA, USA). RNA concentration and integrity were determined using a Nanodrop 2000c spectrophotometer (Thermo Scientific, Wilmington, DE, USA) and Agilent 2100 Bioanalyzer (Agilent Technologies, Santa Clara, CA, USA) as previously described [20,135]. Samples with RNA integrity numbers (RIN) greater than or equal to nine were used for library preparation. One microgram of total RNA from each tissue was used to produce strand-specific cDNA libraries using the Truseq (Illumina, San Diego, CA, USA) according to the manufacturer’s instructions. The RNA sequencing was carried out at Novogene Corporation Inc. The computational analysis was started by the establishment of pairs of Fastq files for all samples. The workflow consisted of QC check (FastQC) -> alignment (Hisat2) -> feature counts (subread) -> Differential gene expression analysis (DESeq2). For quality control, FastQC was used to check the quality of raw fastq data from the sequencing core and after adaptor cut and quality trimming [136]. FastQC reports consisted of multiple parameters such as the number of reads, duplicates, adapter contents, and sequence quality score. HISAT2 was applied to perform alignment [137]. The reads of raw fastq data with either end were distributed from 17.1 M to 34.8 M, and the alignment rates were higher than 95%. Assigned reads for feature count distributed from 17.7 M to 36.7 M in correspondence to the rate of assigned reads from 80.1% to 96.5%. Each sample produced a bam file after alignment to the genome. Features from each bam file that mapped to the genome in the provided annotation file were counted by the subread function [138]. MultiQC was used to analyze and integrate the QC reports with input data from reports of fastqc, alignment reads, and feature assigned [139]. An R package DESeq2 was used to analyze differential gene expression [140]. Markers/genes with the sum of read counts across all cases and controls that were ten or greater were kept for the downstream analysis. During DGE analysis, a boxplot of the Cook’s distances was made to determine if any sample raw reads departed from others. All our samples’ raw reads in the boxplot were relatively even, and all samples were included in the downstream analysis for differential gene expression (DGE). To visualize the strength of differential gene expression, the Hierarchical clustering, TreeView graph, and Pathway Enrichment Analysis plot were made using Flaski [141], and the Protein-Protein Interaction Networks were made by the Search Tool for the Retrieval of Interacting Genes (STRING) database [142]. Overall, all the differential gene expressions were acceptable for subsequent statistical analysis. The RNA sequencing data is deposited in the Gene Expression Omnibus (GEO) database with accession number (GSE224991).

### 4.4. Quantitative RT-PCR

Briefly, 2 μg of RNA was reverse transcribed using random primers for selected genes according to the manufacturer’s guidelines (Applied Biosystems, Carlsbad, CA, USA). Quantitative RT-PCR was carried out using the SYBR gene expression master mix (Applied Biosystems). Reactions were incubated for 10 min at 95 °C followed by 40 cycles for 15 s at 95 °C and 1 min at 60 °C. The expression levels of selected genes were quantified using Invitrogen StepOne System with FBXW2 (F-box and WD repeat domain containing two) used for normalization [143]. All reactions were run in triplicate, and relative mRNA expression was determined using the comparative cycle threshold method (2^−ΔΔCq^), as recommended by the supplier (Applied Biosystems). Values were expressed as fold change compared to the control group. The primer sequences in the 5′–3′ direction used are listed in Appendix A.

### 4.5. Immunoblotting

Total protein isolated from leiomyoma and paired myometrium was subjected to immunoblotting as previously described [110,144]. Briefly, samples were suspended in RIPA buffer containing 1 mM EDTA and EGTA (Boston BioProducts, Ashland, MA, USA) supplemented with 1 mM PMSF and a complete protease inhibitor mixture (Roche Diagnostics, Indianapolis, IN, USA), sonicated, and centrifuged at 4 °C for 10 min at 14,000 rpm. The concentration of protein was determined using the BCA™ Protein Assay Kit (Thermo Scientific Pierce, Rockford, IL, USA). Equal aliquots (thirty micrograms) of total protein for each sample were denatured with SDS-PAGE sample buffer and separated by electrophoresis on an SDS polyacrylamide gel. After transferring the samples to a nitrocellulose membrane, the membrane was blocked with TBS-Tween + 5% milk and probed with PGR primary antibody (Cell Signaling Technology, Danvers, MA, USA). The membranes were washed with TBS containing 0.1%Tween-20 wash buffer after each antibody incubation cycle. SuperSignal West Pico Chemiluminescent Substrate™ (Thermo Scientific Pierce) was used for detection, and photographic emulsion was used to identify the protein bands, which were subsequently quantified by densitometry. The densities of the specific protein bands were determined using the Image J program (http://imagej.nih.gov/ij/ accessed on 7 July 2023), normalized to a band obtained from staining the membrane with Ponceau S. Results were expressed as means ± SEM as a ratio relative to the control group designated as 1.

### 4.6. Statistical Analysis

Throughout the text, results are presented as mean ± SEM and analyzed by GraphPad Prism 10 software (Graph-Pad, San Diego, CA, USA). Dataset normality was determined by the Kolmogorov–Smirnoff test. The data presented in this study were not normally distributed; therefore, non-parametric tests were used. Comparisons involving two groups were analyzed using the Wilcoxon matched-pairs signed rank test (Figure 5B and Appendix A) or the Mann-Whitney test (Appendix A). The Kruskal-Wallis Test was used for comparisons involving multiple groups (Figure 3, Figure 4, Figure 5C, and Appendix A). Statistical significance was established at *p* < 0.05.

## Figures and Tables

**Figure 1 ijms-24-13441-f001:**
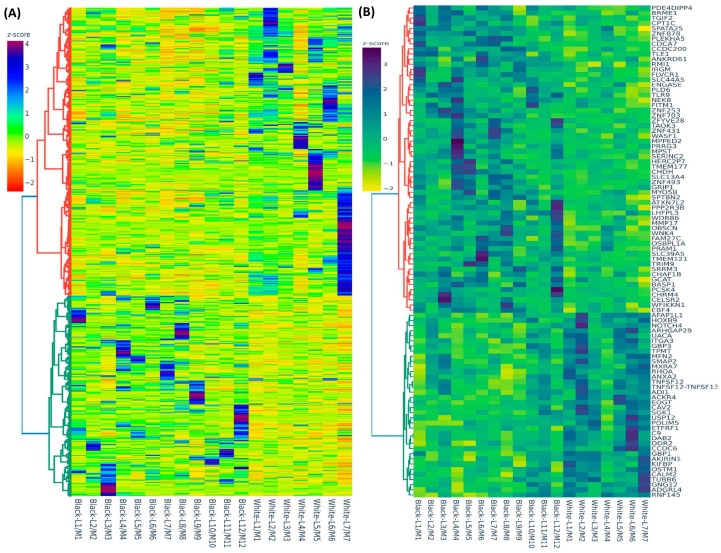
(**A**) Hierarchical clustered heatmap analysis was performed as fold change (Leiomyoma/paired Myometrium) comparing Black (n = 12) with White group (n = 7) (fold change ≥ 1.5, *p* < 0.05). Color gradient represents gene expression as z-scores. (**B**) Heatmap of the 95 enriched transcripts (Leiomyoma/paired Myometrium) in the Black group (n = 12) but not in the White group (n = 7) (fold change ≥ 1.5, *p* < 0.05). Color gradient represents gene expression levels as z-scores.

**Figure 2 ijms-24-13441-f002:**
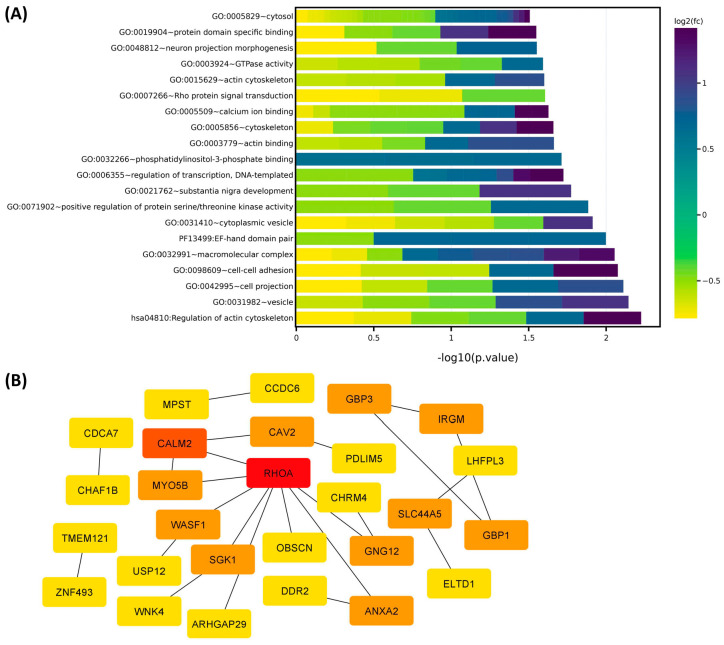
(**A**) Gene ontology (GO) analysis of 95 differentially-expressed transcripts (Leiomyoma/paired Myometrium) in the Black (n = 12) but not in the White group (n = 7) (fold change ≥ 1.5, *p* < 0.05). Color gradient represents levels of log2 fold change presented as z-scores. (**B**) The Protein-Protein Interaction Networks were constructed by the Search Tool for the Retrieval of Interacting Genes (STRING) database and Cytoscape software version 3.9.1 using the 27 hub genes identified by the CytoHubba plugin of Cytoscape software platform. The color of the nodes denotes the degree of interaction among the genes (from high- to low-degree: red, orange, and yellow).

**Figure 3 ijms-24-13441-f003:**
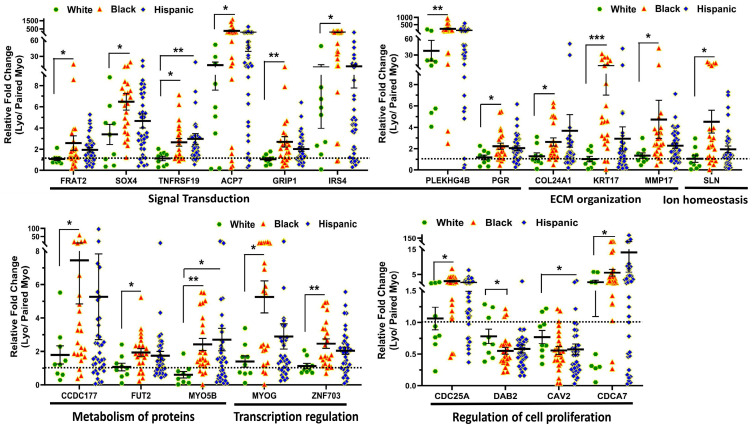
The mRNA expression of FRAT2, SOX4, TNFRSF19, ACP7, GRIP1 and IRS4, PLEKHG4B, PGR, COL24A1, KRT17, MMP17 and SLN, CDC25A, CCDC177, FUT2, MYO5B, MYOG and ZNF703, DAB2, CAV2, and CDCA7 expressed as fold change (Lyo/ paired Myo) in White (n = 9), Black (n = 23), and Hispanic groups (n = 37) by qRT-PCR. The results are presented as mean ± SEM with *p* values (* *p* < 0.05; ** *p* < 0.01; *** *p* < 0.001) as indicated by the corresponding lines.

**Figure 4 ijms-24-13441-f004:**
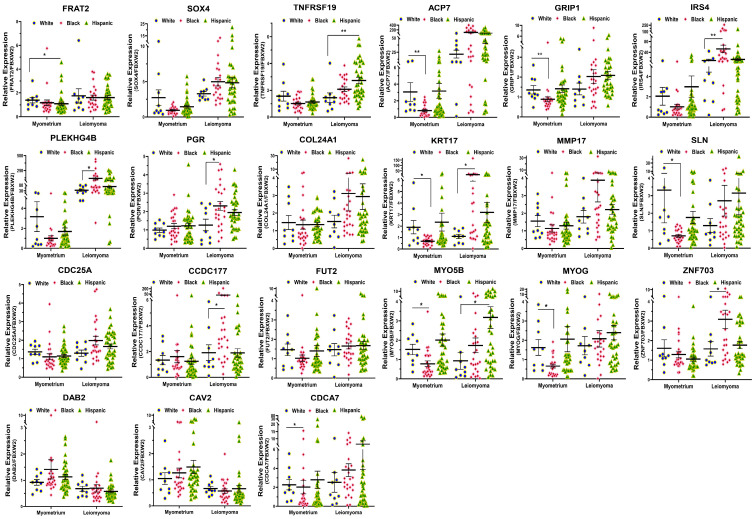
The mRNA expression of FRAT2, SOX4, TNFRSF19, ACP7, GRIP1 and IRS4, PLEKHG4B, PGR, COL24A1, KRT17, MMP17 and SLN, CDC25A, CCDC177, FUT2, MYO5B, MYOG and ZNF703, DAB2, CAV2, and CDCA7 expressed in the myometrium and leiomyomas in White (n = 9), Black (n = 23), and Hispanic groups (n = 37). The results are presented as mean ± SEM with *p* values (* *p* < 0.05; ** *p* < 0.01) as indicated by the corresponding lines.

**Figure 5 ijms-24-13441-f005:**
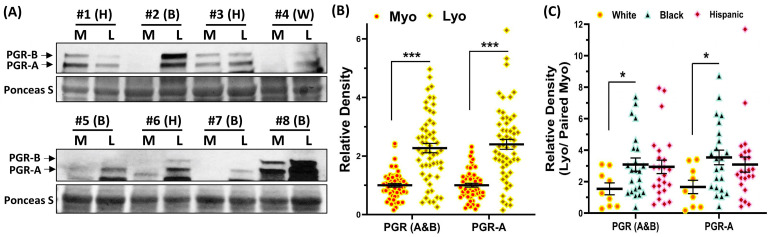
(**A**) Representative Western blot analysis of PGR-A and PGR-B with bar graphs (**B**,**C**) showing their relative band densities in paired myometrium and leiomyomas ((**B**), n = 56) and fold change ((**C**); Lyo/paired Myo) in White (n = 9), Black (n = 23), and Hispanic groups (n =24). The results are presented as mean ± SEM with *p* values (* *p* < 0.05; *** *p* < 0.001) as indicated by the corresponding lines.

**Table 1 ijms-24-13441-t001:** The list of 95 coding transcripts only altered significantly in the Black group.

ENSEMBL_GENE_ID	Genes	log_2_ (FoldChange)	*p*adj
ENSG00000129048	ACKR4	−0.98571759418595	0.016261611
ENSG00000162618	ADGRL4	−0.730451745246549	0.021985159
ENSG00000182551	ADI1	−0.473711465167378	0.0481993
ENSG00000157510	AFAP1L1	−0.904978358197408	0.01851514
ENSG00000174574	AKIRIN1	−0.590414814282603	0.003694255
ENSG00000157999	ANKRD61	0.691198791	0.034099101
ENSG00000182718	ANXA2	−0.618428859469741	0.019572457
ENSG00000137962	ARHGAP29	−0.718256380610321	0.042451786
ENSG00000162650	ATXN7L2	0.678347659	0.042636838
ENSG00000176788	BASP1	1.07947171	0.006518371
ENSG00000132016	BRME1	0.661505175	0.020851774
ENSG00000113600	C9	−0.703595240369084	0.026897711
ENSG00000143933	CALM2	−0.489335184417676	0.008440807
ENSG00000105971	CAV2	−0.851491859685069	0.038728692
ENSG00000236383	CCDC200	0.605100694	0.012214075
ENSG00000108091	CCDC6	−0.72039613537964	0.04919997
ENSG00000144354	CDCA7	3.131437732	0.005090236
ENSG00000143126	CELSR2	1.420142912	0.044457001
ENSG00000159259	CHAF1B	0.78736468	0.022420806
ENSG00000016391	CHDH	1.855973134	0.004550716
ENSG00000180720	CHRM4	1.601579406	0.025196859
ENSG00000169169	CPT1C	0.909847927	0.023118876
ENSG00000153071	DAB2	−0.704920969888342	0.026402222
ENSG00000162733	DDR2	−0.554606158437811	0.009311408
ENSG00000088881	EBF4	0.703879433	0.003016076
ENSG00000167280	ENGASE	0.59590162	0.029004618
ENSG00000163378	EOGT	−0.883114504762434	0.012720403
ENSG00000205707	ETFRF1	−0.408436567922212	0.032513346
ENSG00000231527	FAM27C	0.797595443	0.013815831
ENSG00000285321	FITM1	1.249462295	0.042681194
ENSG00000162769	FLVCR1	0.594590046	0.006516276
ENSG00000117228	GBP1	−0.580760531241074	0.005806013
ENSG00000117226	GBP3	−0.639124945111892	0.046454847
ENSG00000100116	GCAT	0.682650655	0.031571242
ENSG00000172380	GNG12	−0.473486408045602	0.025014425
ENSG00000155974	GRIP1	1.084594681	0.019454493
ENSG00000170689	HOXB9	−1.60364025325228	0.016884729
ENSG00000237693	IRGM	0.684838416	0.008265492
ENSG00000005884	ITGA3	−0.785456415144709	0.024192267
ENSG00000198954	KIFBP	−0.404000966500028	0.041977871
ENSG00000187416	LHFPL3	0.900676029	0.023206141
ENSG00000116688	MFN2	−0.437884181348915	0.036758025
ENSG00000198598	MMP17	1.282282893	0.00061147
ENSG00000066382	MPPED2	1.950018885	0.045996611
ENSG00000128309	MPST	0.783557462	0.024797843
ENSG00000182534	MXRA7	−0.455181924885382	0.030824687
ENSG00000167306	MYO5B	0.856051914	0.041759626
ENSG00000160602	NEK8	0.627987206	0.047865902
ENSG00000235396	NOTCH4	−0.484552663548917	0.041998944
ENSG00000154358	OBSCN	0.656724611	0.035896745
ENSG00000141447	OSBPL1A	0.693232152	0.011515721
ENSG00000081087	OSTM1	−0.550742788524356	0.020084338
ENSG00000115257	PCSK4	1.30636991	0.04361426
ENSG00000290904	PDE4DIPP4	0.822554506	0.026796414
ENSG00000163110	PDLIM5	−0.616541920543817	0.02817133
ENSG00000179598	PLD6	0.763237458	0.008882134
ENSG00000052126	PLEKHA5	0.625224515	0.011933448
ENSG00000276438	PPP2R3B	0.68893296	0.040025832
ENSG00000133246	PRAM1	0.84169506	0.040427487
ENSG00000130032	PRRG3	1.418129302	0.033504092
ENSG00000067560	RHOA	−0.414796410616333	0.0240862
ENSG00000178966	RMI1	0.828504136	0.001401385
ENSG00000145860	RNF145	−0.398685120823013	0.013611116
ENSG00000168528	SERINC2	1.25222147	0.02318918
ENSG00000118515	SGK1	−1.1217221618793	0.037292795
ENSG00000164707	SLC13A4	0.625492215	0.015016236
ENSG00000139540	SLC39A5	0.889453589	0.048991739
ENSG00000137968	SLC44A5	1.174040295	0.010752015
ENSG00000084070	SMAP2	−0.471202785875124	0.029158884
ENSG00000149634	SPATA25	0.844104476	0.027233962
ENSG00000173898	SPTBN2	0.857422484	0.005325753
ENSG00000177679	SRRM3	0.956168836	0.036233089
ENSG00000135090	TAOK3	0.722318639	0.020904922
ENSG00000118707	TGIF2	0.684452361	0.027880253
ENSG00000196781	TLE1	0.604094935	0.006045956
ENSG00000239732	TLR9	0.994924051	0.010095372
ENSG00000184986	TMEM121	1.255528818	0.012071561
ENSG00000144120	TMEM177	0.649576577	0.025741594
ENSG00000239697	TNFSF12	−0.504945190909519	0.042070126
ENSG00000248871	TNFSF12-TNFSF13	−0.524933065913806	0.032530573
ENSG00000137364	TPMT	−0.599933728185664	0.036726324
ENSG00000100505	TRIM9	1.963547604	0.021114898
ENSG00000176014	TUBB6	−0.583604412902083	0.04341609
ENSG00000137831	UACA	−0.459808559154032	0.041008333
ENSG00000152484	USP12	−0.667776945538996	0.036226476
ENSG00000112290	WASF1	0.661083104	0.014216308
ENSG00000187260	WDR86	0.782512036	0.016809294
ENSG00000127578	WFIKKN1	0.822204726	0.022154194
ENSG00000126562	WNK4	1.21048519	0.012389336
ENSG00000159733	ZFYVE28	0.649531661	0.045798091
ENSG00000256771	ZNF253	0.891216558	0.008172245
ENSG00000196705	ZNF431	0.592983837	0.020198178
ENSG00000196268	ZNF493	0.687387776	0.01036717
ENSG00000183779	ZNF703	1.320402356	0.007772286
ENSG00000257446	ZNF878	1.055383649	0.001750249

**Table 2 ijms-24-13441-t002:** Genes selected based on the RNAseq analysis according to race/ethnicity.

GO/KEGG Pathway Enrichment	Symbol	Lyo vs. Myo	Black^(Lyo/Myo)^ vs. White^(Lyo/Myo)^	Black^(Lyo)^ vs. White^(Lyo)^	Black^(Myo)^ vs. White^(Myo)^	MED12-Mut^(Lyo/Myo)^ vs. MED12-WT^(Lyo/Myo)^	Function
Signal Transduction	FRAT2	Up (*p* < 0.001)	Up (*p* < 0.05)	No Significance	No Significance	Up (*p* < 0.05)	Positive WNT signaling pathway regulator; promotes the abnormal β-catenin accumulation through sequestering GSK-3β protein, by which carcinogenesis is promoted [31].
Signal Transduction	SOX4	Up (*p* < 0.001)	Up (*p* < 0.05)	No Significance	No Significance	No Significance	DNA-binding transcription factor; related to the apoptosis pathway and WNT and ERK signaling pathways, as well as tumorigenesis and involved in the regulation of bone and embryonic development and cell fate determination [32].
Signal Transduction	TNFRSF19	Up (*p* < 0.001)	Up (*p* < 0.05)	No Significance	No Significance	Up (*p* < 0.05)	Member of the TNF-receptor superfamily; essential in embryonic development, and involved in regulation of JNK, WNT/β-catenin and TGF-β signaling pathway, and apoptosis in a caspase-independent mechanism [33,34].
Signal Transduction	ACP7	Up (*p* < 0.001)	Up (*p* < 0.05)	No Significance	Down (*p* < 0.01)	Up (*p* < 0.01)	Member of purple acid phosphatases family, plays a critical role in the prognosis of glioblastoma [35,36].
Signal Transduction	GRIP1	Up (*p* < 0.001)	Up (*p* < 0.01)	No Significance	Down (*p* < 0.01)	No Significance	Member of the glutamate receptor interacting protein family, assembles multiprotein signaling complex and mediates the membrane organization through trafficking of its binding partners to specific location [37].
Signal Transduction	IRS4	Up (*p* < 0.001)	Up (*p* < 0.05)	Up (*p* < 0.01)	No Significance	Up (*p* < 0.01)	Contains many potential tyrosine and serine/threonine phosphorylation sites and has been shown to contribute to tumor initiation and progression through mediating a complex network of cytoplasmic signaling upon receptor stimulation [38].
Signal Transduction	PLEKHG4B	Up (*p* < 0.001)	Up (*p* < 0.01)	Up (*p* < 0.05)	No Significance	Up (*p* < 0.001)	Member of Rho-guanine nucleotide exchange factor, which contains a pleckstrin homology domain and plays an essential role in activation of Cdc42 [39].
Signal Transduction	PGR	Up (*p* < 0.001)	Up (*p* < 0.05)	Up (*p* < 0.05)	No Significance	No Significance	PGR includes two isoforms (A and B), which plays a central role in the physiological effects of progesterone acting as transcriptional activator or repressor [40].
Extracellular matrix organization	COL24A1	Up (*p* < 0.001)	Up (*p* < 0.05)	No Significance	No Significance	No Significance	Member of collagen gene family; participates in regulation of type I collagen fibrillogenesis during fetal development at specific anatomical locations [41].
Extracellular matrix organization	KRT17	Up (*p* < 0.001)	Up (*p* < 0.001)	Up (*p* < 0.05)	Down (*p* < 0.05)	Up (*p* < 0.05)	A multifunctional protein that plays an important role in the formation and maintenance of cellular structural support; regulates a variety of biological processes such as cell proliferation, apoptosis, migration, and signal transduction [42].
Extracellular matrix organization	MMP17	Up (*p* < 0.001)	Up (*p* < 0.05)	No Significance	No Significance	No Significance	Involved in the breakdown of various components of extracellular matrix and precursors of transmembrane inflammatory mediators or growth factors; overexpressed in multiple cancers and aasociated with tumor progression [43].
Ion homeostasis	SLN	Up (*p* < 0.01)	Up (*p* < 0.05)	No Significance	Down (*p* < 0.05)	No Significance	Is one of the sarcoplasmic reticulum Ca(2+)-ATPases; plays an essential role in calcium homeostasis through modulation of calcium re-uptake in muscle cells [44].
Metabolism of proteins	CCDC177	Up (*p* < 0.001)	Up (*p* < 0.05)	Up (*p* < 0.05)	No Significance	No Significance	Associated with myopathy, fiber-type disproportion, and Pontiac fever. Has a significant prognostic value in prediction of overall survival in lung squamous cell carcinoma patients [45].
Metabolism of proteins	FUT2	Up (*p* < 0.001)	Up (*p* < 0.05)	No Significance	No Significance	No Significance	is a galactoside 2-L-fucosyltransferase enzyme; overexpressed and involved in regulation of proliferation and metastasis of colorectal cancer through activation of Wnt/β-catenin pathway [46].
Metabolism of proteins	MYO5B	Up (*p* < 0.01)	Up (*p* < 0.01)	No Significance	Down (*p* < 0.05)	No Significance	MYO5B is involved in plasma membrane recycling and vesicular trafficking. It has been associated with bipolar disorder, and its mutation caused microvillous inclusion disease because of deficient trafficking of basolateral and apical proteins [47].
Transcription regulation	MYOG	Up (*p* < 0.001)	Up (*p* < 0.05)	No Significance	Down (*p* < 0.05)	No Significance	Is a muscle-specific transcriptional activator that promotes myogenesis, muscle differentiation, and involved in preventing reversal of muscle cell differentiation. It induces fibroblasts to differentiate into myoblasts and is essential for the development of functional embryonic skeletal muscle [48].
Transcription regulation	ZNF703	Up (*p* < 0.001)	Up (*p* < 0.01)	Up (*p* < 0.05)	No Significance	Up (*p* < 0.05)	Identified as an oncogene and involved in regulation of cell adhesion, migration, proliferation, and gene transcription through recruitment of histone deacetylases [49].
Regulation of cell proliferation	CDC25A	Up (*p* < 0.001)	Up (*p* < 0.05)	No Significance	No Significance	No Significance	Member of the tyrosine protein phosphatase; regulates cell cycle progression from G1 to the S phase by targeting cyclin-dependent kinase CDC2, CDK1 and CDK2. CDC25A is considered as an oncogene through its involvement in RAS, E2F family and p53-p21-Cdk axis mediated carcinogenesis [50].
Regulation of cell proliferation	DAB2	Down (*p* < 0.001)	Down (*p* < 0.05)	No Significance	No Significance	No Significance	A mitogen-responsive phosphoprotein; is a key regulator of anti- or pro-tumorigenic pathways [51].
Regulation of cell proliferation	CAV2	Down (*p* < 0.001)	Down (*p* < 0.05)	No Significance	No Significance	Down (*p* < 0.05)	A scaffolding protein within caveolar membranes; involved in insulin-stimulated nuclear translocation and essential cellular functions such as cellular growth control, lipid metabolism, and signal transduction including MAPK, STAT3, EGFR and G protein α-associated signaling pathway [52].
Regulation of cell proliferation	CDCA7	Up (*p* < 0.01)	Up (*p* < 0.05)	No Significance	Down (*p* < 0.05)	No Significance	Is a validated c-Myc responsive gene and involved in c-MYC-mediated cell transformation, tumorigenesis, and anchorage-independent growth and clonogenicity of lymphoblastoid cells [53].

## Data Availability

Raw data were generated at The Lundquist Institute. Derived data supporting the findings of this study are available from the corresponding author, O.K., on request.

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
