# Peer review of "The Influence of Race/Ethnicity on the Transcriptomic Landscape of Uterine Fibroids"

_ijms, 2023, doi:10.3390/ijms241713441_

Round 1

Reviewer 1 Report

Chuang et al. performed RNA-seq in Fibroid/paired Myo tissues from Black, White, and Hispanic women to identify genes that could explain the observed racial disparity in fibroid disease.

Introduction. The authors present results on samples obtained from Hispanic women. Nevertheless, the introduction lacks information about this specific population. The introduction must improve.

“The analysis indicated that 95 genes were minimally changed in tumors from White (DF≈1) but were significantly altered by 17 more than 1.5-fold (up or down) in Black patients”. Provide further detail regarding the criteria that led to the selection of those particular 21 genes for validation.

M&M. Authors mentioned that “The 19 pairs of tissues used for next-generation RNA sequencing were from 11 MED12 mutation-positive and 8 MED12 mutation-negative leiomyomas. The mutation analysis of the specimens (n=69) indicated that 46 leiomyomas had the MED12 mutations (46/69 pairs; 66.7%) with no mutations in the myometrium” (lines 401-404). How the mutation % is distributed between races? How this can affect the results observed? For example, if the majority of MED12 mutation samples are found within the Black group, how can the authors confidently attribute the observation to race rather than to the MED12 mutation, as they have previously demonstrated?

Results. It remains unclear why authors present results intermittently based on race, MED12 mutation, or by comparing adjacent myometrium to fibroids. A significant reorganization of the manuscript is warranted.

The Hispanic group is depicted in all the graphs, yet the authors do not note distinctions in the results section between this group and other groups (White and Black). Furthermore, the authors stated, "CAV2 was significantly lower in Black group as compared to White group (Fig. 3)" (Line 147), but the asterisk on the graph indicates differences between the White and Hispanic groups. This requires clarification.

Other comments:

Title: "Race and Ethnicity" could be confusing since the authors did not mention any difference between these terms. I suggest adding a dash between the words.

Figure legend 1 B: Replace "96" with "95."

Figure 3. I recommend categorizing the results according to signaling pathways. For example, under "ECM organization" include COL24A1, KRT17, and MMP17.

Add a dashed line to the figures displaying "Relative FC (Lyo/paired Myo)" to facilitate the identification of minimally or non-altered expressions in the groups.

Figure 5. Specify the race of each patient in the representative gel. Elaborate on the meaning of the "PS" abbreviation in the figure legend.

Author Response

Reviewer #1
1.
Chuang et al. performed RNA-seq in Fibroid/paired Myo tissues from Black, White, and Hispanic women to identify genes that could explain the observed racial disparity in fibroid disease.

Introduction. The authors present results on samples obtained from Hispanic women. Nevertheless, the introduction lacks information about this specific population. The introduction must improve.

Response: Thanks for the suggestions. We have updated the introduction to reflect the limited Hispanic-specific findings in leiomyoma associated research.

  1. “The analysis indicated that 95 genes were minimally changed in tumors from White (DF≈1) but were significantly altered by more than 1.5-fold (up or down) in Black patients”. Provide further detail regarding the criteria that led to the selection of those particular 21 genes for validation.

Response: Thanks for the suggestions. We have included a sentence in the results section:

“Among the 21 genes for validation, 7 (CAV2, CDCA7, DAB2, GRIP1, MMP17, MYO5B and ZNF703) were selected from the 95 genes that were altered significantly in tumors from the Black group, but minimally changed in White patients. The remaining 14 genes were selected from our RNAseq analysis because of their significant race dependence and their functional involvement in reproduction.”  

  1. M&M. Authors mentioned that “The 19 pairs of tissues used for next-generation RNA sequencing were from 11 MED12 mutation-positive and 8 MED12 mutation-negative leiomyomas. The mutation analysis of the specimens (n=69) indicated that 46 leiomyomas had the MED12 mutations (46/69 pairs; 66.7%) with no mutations in the myometrium” (lines 401-404). How the mutation % is distributed between races? How this can affect the results observed? For example, if the majority of MED12 mutation samples are found within the Black group, how can the authors confidently attribute the observation to race rather than to the MED12 mutation, as they have previously demonstrated?

Response: Thanks for the suggestions. The MED12 mutation rate in White patients was 55.6%, 76% in Black and 62.9% in the Hispanic group. Although there was a slightly higher MED12 mutation rate in the Black group, among the 9 genes (FRAT2, TNFRSF19, GRIP1, PGR, KRT17, SLN, CDC25A, FUT2 and ZNF703) that were minimally or not altered in the White group but were expressed significantly more in the Black group, the expression of only 4 genes (FRAT2, TNFRSF19, KRT17 and ZNF703) correlated with MED12 mutation. In addition, among the 21 genes that were confirmed to exhibit racial differences in their expression, only 8 genes had correlation with MED12 mutation status. Furthermore, several reports (PMID: 22182697; 22428002; 25325994; 30099503) have indicated that the MED12 mutation rate in leiomyomas regardless of race/ethnicity is about the same. Therefore, we consider race as an important factor leading to differential gene expression; however, we could not rule out the impact of MED12 mutation status in our racial analysis because of our limited number of specimens in each race/ethnicity group.    

  1. Results. It remains unclear why authors present results intermittently based on race, MED12 mutation, or by comparing adjacent myometrium to fibroids. A significant reorganization of the manuscript is warranted.

Response: Thanks for the suggestions. As we mentioned in the introduction section our objective in this gene expression profiling study was to determine if there are genes that are uniquely expressed in a race-dependent manner. Previous report from our group (PMID: 36835153) focused on the association between the MED12 mutation status and gene expression in fibroids, we therefore performed the confirmation analysis based on MED12 mutation status (Supplementary Fig. 4). In addition, because a recent study (PMID: 36066972) indicated that the myometrial compartment could be the potential source for fibroid transformation, we performed analysis of gene expression in the matched adjacent myometrium.

  1. The Hispanic group is depicted in all the graphs, yet the authors do not note distinctions in the results section between this group and other groups (White and Black). Furthermore, the authors stated, "CAV2 was significantly lower in Black group as compared to White group (Fig. 3)" (Line 147), but the asterisk on the graph indicates differences between the White and Hispanic groups. This requires clarification.

Response: Thanks for the suggestions. We have edited the sentence as below:

“……while the expression of DAB2 was significantly lower in Black group as compared to the White group. In addition,  the expression of CAV2 mRNA was significantly higher in tumors from Hispanic patients as compared to tumors from White patients (Fig. 3).”

Other comments:

Title: "Race and Ethnicity" could be confusing since the authors did not mention any difference between these terms. I suggest adding a dash between the words.

Response: Thanks for the suggestion. We have revised as suggested.

Figure legend 1 B: Replace "96" with "95."

Response: Thanks for the suggestions. It has been corrected.

Figure 3. I recommend categorizing the results according to signaling pathways. For example, under "ECM organization" include COL24A1, KRT17, and MMP17.

Response: Thanks for the suggestion. We have revised as suggested.

Add a dashed line to the figures displaying "Relative FC (Lyo/paired Myo)" to facilitate the identification of minimally or non-altered expressions in the groups.

Response: Thanks for the suggestions. We have revised accordingly.

Figure 5. Specify the race of each patient in the representative gel. Elaborate on the meaning of the "PS" abbreviation in the figure legend.

Response: Thanks for the suggestion. We have revised accordingly.

Reviewer 2 Report

The manuscript " The influence of Race and Ethnicity on The Transcriptomic Landscape of Uterine Fibroids" is a detailed research paper and well referenced paper.  The manuscript will be of interest to both Gynecologists and Histopathologists.

Some suggestions to consider are listed below:

As the Results and Discussion are listed before the Materials and Methods the paper is difficult to read.

In the Material and Methods section of the manuscript " Portions of uterine leiomyomas (intramural, 3-5 cm in diameters) with adjacent myometrium were sampled. "  As also stated in the Introduction -Line 48 " Genome wide association studies have revealed an association of a number of genes with fibroid size and volume in a race dependent manner [17, 18]."  Could additional information be added in regards to the decision to choose only intramural fibroids as opposed to submucosal/subserosal and 3-5 cm diameter fibroids.  In addition was there confirmation of a leiomyoma after processing and histological examination of the sampled macroscopically appearing fibroid?

Author Response

Reviewer #2
1.
In the Material and Methods section of the manuscript " Portions of uterine leiomyomas (intramural, 3-5 cm in diameters) with adjacent myometrium were sampled. "  As also stated in the Introduction -Line 48 " Genome wide association studies have revealed an association of a number of genes with fibroid size and volume in a race dependent manner [17, 18]."  Could additional information be added in regards to the decision to choose only intramural fibroids as opposed to submucosal/subserosal and 3-5 cm diameter fibroids.  In addition was there confirmation of a leiomyoma after processing and histological examination of the sampled macroscopically appearing fibroid?

Response: Thanks for the suggestions. The reason for limiting the size parameter for fibroids was to reduce the heterogeneity and variance among tissues. This rationale has been included in the Material and Methods section.

“To reduce the variance among fibroids, tumors between 3 to 5 cm in diameter and with intramural location (n=69) were obtained from patients at Harbor-UCLA Medical Center undergoing hysterectomy.”

Reviewer 3 Report

Comments and Suggestions for Authors

Dear Authors,

I reviewed with interest the paper entitled “The influence of Race and Ethnicity on The Transcriptomic Landscape of Uterine Fibroids”.

First, I would strongly congratulate with the authors for their hard work, which covers an actual and very interesting topic.

I found the present study interesting and fluent to read.

Methods are well described and presented. The figures are of good quality.

Results are clearly written.

Minor revision:

Please put the Materials and Methods section after the introduction and not at the end of the article.

Racial disparity and its associated symptoms is a hot topic when exploring benign and malignant pathologies. Please also refer to:  The Impact of Ethnicity and Age on Distribution of Metastases in Patients with Upper Tract Urothelial Carcinoma: Analysis of SEER Data. Biomedicines 202311, 1943. https://doi.org/10.3390/biomedicines11071943.

Author Response

Reviewer #3

  1. Racial disparity and its associated symptoms is a hot topic when exploring benign and malignant pathologies. Please also refer to: The Impact of Ethnicity and Age on Distribution of Metastases in Patients with Upper Tract Urothelial Carcinoma: Analysis of SEER Data. Biomedicines 2023, 11, 1943. https://doi.org/10.3390/biomedicines11071943.

Response: Thanks for the suggestion. This reference has been included.

Round 2

Reviewer 1 Report

The authors have addressed most of my comments.

However, I suggest adding MED12 mutation % information in the M&M section (i.e.: "The MED12 mutation rate in White patients was 55.6%, 76% in Black and 62.9% in the Hispanic group") and the rest of their answer to the discussion section ("Although there was a slightly higher MED12 mutation rate in the Black group, among the 9 genes (FRAT2, TNFRSF19, GRIP1, PGR, KRT17, SLN, CDC25A, FUT2 and ZNF703) that were minimally or not altered in the White group but were expressed significantly more in the Black group, the expression of only 4 genes (FRAT2, TNFRSF19, KRT17 and ZNF703) correlated with MED12 mutation. In addition, among the 21 genes that were confirmed to exhibit racial differences in their expression, only 8 genes had correlation with MED12 mutation status. Furthermore, several reports (PMID: 22182697; 22428002; 25325994; 30099503) have indicated that the MED12 mutation rate in leiomyomas regardless of race/ethnicity is about the same. Therefore, we consider race as an important factor leading to differential gene expression; however, we could not rule out the impact of MED12 mutation status in our racial analysis because of our limited number of specimens in each race/ethnicity group"). Modify accordingly if necessary.

Author Response

Reviewer #1
1.
The authors have addressed most of my comments.

However, I suggest adding MED12 mutation % information in the M&M section (i.e.: "The MED12 mutation rate in White patients was 55.6%, 73.9% in Black and 62.2% in the Hispanic group") and the rest of their answer to the discussion section ("Although there was a slightly higher MED12 mutation rate in the Black group, among the 9 genes (FRAT2, TNFRSF19, GRIP1, PGR, KRT17, SLN, CDC25A, FUT2 and ZNF703) that were minimally or not altered in the White group but were expressed significantly more in the Black group, the expression of only 4 genes (FRAT2, TNFRSF19, KRT17 and ZNF703) correlated with MED12 mutation. In addition, among the 21 genes that were confirmed to exhibit racial differences in their expression, only 8 genes had correlation with MED12 mutation status. Furthermore, several reports (PMID: 22182697; 22428002; 25325994; 30099503) have indicated that the MED12 mutation rate in leiomyomas regardless of race/ethnicity is about the same. Therefore, we consider race as an important factor leading to differential gene expression; however, we could not rule out the impact of MED12 mutation status in our racial analysis because of our limited number of specimens in each race/ethnicity group"). Modify accordingly if necessary.

Response: Thanks for the suggestion. We have revised accordingly.